# The *Lactobacillus gasseri* G098 Strain Mitigates Symptoms of DSS-Induced Inflammatory Bowel Disease in Mice

**DOI:** 10.3390/nu14183745

**Published:** 2022-09-10

**Authors:** Wei-Qin Zhang, Ke-Yu Quan, Cui-Jiao Feng, Tao Zhang, Qiu-Wen He, Lai-Yu Kwok, Yong-Fu Chen

**Affiliations:** 1Key Laboratory of Dairy Biotechnology and Engineering, Ministry of Education, Inner Mongolia Agricultural University, Hohhot 010018, China; 2Key Laboratory of Dairy Products Processing, Ministry of Agriculture and Rural Affairs, Inner Mongolia Agricultural University, Hohhot 010018, China; 3Inner Mongolia Key Laboratory of Dairy Biotechnology and Engineering, Inner Mongolia Agricultural University, Hohhot 010018, China

**Keywords:** inflammatory bowel disease, *Lactobacillus gasseri*, gut microbiome, colonic tissues

## Abstract

Inflammatory bowel disease (IBD) is a recurring inflammatory disease of the gastrointestinal tract with unclear etiology, but it is thought to be related to factors like immune abnormalities and gut microbial dysbiosis. Probiotics can regulate host immunity and gut microbiota; thus, we investigated the alleviation effect and mechanism of the strain *Lactobacillus gasseri* G098 (G098) on dextran sodium sulfate (DSS)-induced colitis in mice. Three groups of mice (*n* = 8 per group) were included: normal control (NC), DSS-induced colitis mice (DSS), and colitis mice given strain (G098). Our results showed that administering G098 effectively reversed DSS-induced colitis-associated symptoms (mitigating weight loss, reducing disease activity index and pathology scores; *p* < 0.05 in all cases) and prevented DSS-induced mortality (62.5% in DSS group; 100% in G098 group). The mortality rate and symptom improvement by G098 administration was accompanied by a healthier serum cytokine balance (significant decreases in serum pro-inflammatory factors, interleukin (IL)-6 [*p* < 0.05], IL-1β [*p* < 0.01], and tumor necrosis factor (TNF)-α [*p* < 0.001], and significant increase in the serum anti-inflammatory factor IL-13 [*p* < 0.01], compared with DSS group) and gut microbiome modulation (characterized by a higher gut microbiota diversity [*p* < 0.05], significantly more *Firmicutes* and *Lachnoclostridium* [*p* < 0.05], significantly fewer *Bacteroidetes* [*p* < 0.05], and significant higher gene abundances of sugar degradation-related pathways [*p* < 0.05], compared with DSS-treated group). Taken altogether, our results suggested that G098 intake could mitigate DSS-induced colitis through modulating host immunity and gut microbiome, and strain treatment is a promising strategy for managing IBD.

## 1. Introduction

Inflammatory bowel disease (IBD), including ulcerative colitis (UC) and Crohn’s disease (CD), is a chronic, non-specific immune-mediated, and recurrent gastrointestinal disease that occurs in the ileum, rectum, and colon. The clinical symptoms of IBD are diarrhea, abdominal pain, and even bloody stool [1]. From a few sporadic cases of IBD in the late 18th century to millions of IBD patients worldwide in the 21st century, the prevalence of IBD has increased explosively over the past 250 years, and the number of patients with IBD worldwide is likely to exceed 10 million in the next decade [2,3]. It has become a global disease, bringing a heavy burden to society and individuals [3]. The pathogenesis of IBD is complicated, and its occurrence and development are affected by many factors, such as genetic susceptibility, immune regulation, barrier integrity, and gut microbiota [4].

The gut microbiota has a key role in human health, providing nutrition and energy to the host, helping the host defend against pathogens, and regulating the intestinal immune system [5]. The destruction of gut microflora homeostasis is closely related to a variety of diseases, including diabetes [6], depression [7], and obesity [8] and likely plays role in promoting IBD. Gut dysbiosis is characterized by a decrease in beneficial bacteria and an increase in pathogenic bacteria, resulting in damage to the intestinal mucosal barrier and an increase in intestinal permeability [9]. Chang et al. (2020) found that patients with IBD had a lower gut microbiota diversity and richness compared with healthy people [10]. The dysbiotic state of the host gut microbiota will also upset the balance of the intestinal immunity system, activating abnormal immune responses [11]. A recent study observed a decrease in CD8 memory T cell responses but an increase in memory B cell responses in patients with IBD [12]. Additionally, the intestinal biopsies of patients with IBD showed a reduction in the expression of the tight junction protein, ZO-1, on both transcriptional and translational levels [13].

In order to alleviate or even cure IBD, researchers are constantly searching for different drug therapeutics, ranging from traditional therapeutic drugs (e.g., mesalazine, corticosteroids, immunosuppressants) to biologic agents (e.g., infliximab) [14] and small molecule biologics (e.g., Janus kinase inhibitors [15]). These are still the mainstream drug treatments of IBD aimed at relieving inflammation [16]. However, patients’ responsiveness towards these drug therapies is limited [17], so there is an urgent need to identify novel strategies in managing IBD. Moreover, many of these drugs are costly and have serious side effects. Therefore, new treatment methods like microecological agents and stem cell transplantation have drawn much attention [18,19], and finding a safer and more effective treatment has become one of the research hotspots of IBD management.

Probiotics are live microorganisms that colonize the human body and confer health benefits to the host. In the gut, probiotics may regulate the host’s mucosal immune function and restore a healthier intestinal microflora [20]. Previous studies have shown that probiotics could alleviate the symptoms of IBD, mainly by restoring a more balanced gut microbiota, regulating host immunity, and enhancing intestinal mucosal function. For example, a recent study showed that two probiotic *Lactobacillus acidophilus* strains could significantly reduce the disease activity index score in an in vivo dextran sodium sulfate (DSS)-induced colitis mouse model, restore the gut microbiota diversity, and up-regulate the anti-inflammatory cytokine interleukin (IL)-10 [21]. Using the same animal model, Jang et al. (2019) showed that supplementing two *Lactobacillus fermentum* strains (KBL374 and KBL375) significantly decreased the levels of pro-inflammatory factors, including IL-4 and interferon-γ, while increasing the level of anti-inflammatory factor IL-10. In addition, *L. fermentum* KBL375 administration increased the gut microbiota diversity and the abundance of beneficial microorganisms, such as *Lactobacillus* and *Akkermansia* [22]. The results of a meta-analysis found that *VSL#*3 was beneficial for maintaining remission in patients with IBD [23]. These results supported that probiotics have great potential in preventing or managing IBD.

*Lactobacillus gasseri* was first discovered and described by Lauer and Kandler in 1980, and it has been shown to maintain intestinal health and regulate host immunity [24,25,26]. However, little research has been reported on the mitigation of ulcerative colitis by *Lactobacillus gasseri*. *Lactobacillus gasseri* G098 (G098) is a strain previously isolated from the intestine of a healthy infant in Lhasa, Tibet Autonomous Region, China, which is preserved in the China General Microbiological Culture Collection Center (CGMCC strain number 22909). The strain shows excellent tolerance to artificial gastrointestinal fluids and bile salt, which is a potentially probiotic strain of *Lactobacillus gasseri*.

The study aimed to investigate if and how the *Lactobacillus gasseri* G098 strain improved the gut ecosystem and alleviated symptoms associated with DSS-induced colitis in a mouse IBD model, with a focus on modulation of inflammatory cytokines and gut microbiota diversity and composition. 

## 2. Materials and Methods

### 2.1. Animals and Experimental Strain

The study was approved by the Ethics Committee of Inner Mongolia Agricultural University (NND2022099) on 12 August 2022. Twenty-four male C57BL/6J mice were obtained from Beijing Huafukang Biotechnology Co., Ltd (HFK, Beijing, China), where they had free access to food and water and were maintained under specific pathogen-free conditions, with a 12-h light-dark cycle at a temperature of 22 ± 2 °C and humidity of 45 ± 10%. All animal experiments were performed as per the guidelines of the Experimental Animal Care and Ethics Committee of Inner Mongolia Agricultural University.

The experimental strain, *Lactobacillus gasseri* G098, was provided by the Key Laboratory of Dairy Biotechnology and Engineering, Ministry of Education, Inner Mongolia Agricultural University, China.

### 2.2. Experimental Design

After a 1-week acclimatization, the mice were randomized into three groups (*n* = 8 per group): NC group, DSS group, and G098 group. From day 0 to day 7, mice in the NC group were provided with distilled water ad libitum, while other groups received DSS (2.5%, *w*/*w*; MW of DSS, 36,000 to 50,000 Da; MP Biomedicals, LLC, Santa Ana, CA, USA) in drinking water to induce IBD. From day 8 to day 17, the NC group and DSS group received 0.9% saline (0.2 mL/mouse/day) by intragastric gavage, while the G098 group received intragastric gavage of *Lactobacillus gasseri* G098 (resuspended in 0.9% saline, 4 × 10^9^ CFU/0.2 mL/mouse/day). During the whole experiment, the body weight, health state (diarrhea, hematochezia, activity), and death of mice were observed and recorded every day (Figure 1). 

Disease activity index (DAI) is used to assess the severity of colitis, which was the sum score of weight loss, diarrhea, and hematochezia. On day 18, fresh feces were collected and stored at −80 °C, and the mice were sacrificed by volatile isoflurane administration. The contents of the colon were removed, and then the colon length and weight were measured. Afterwards, the colon was divided into two parts: the distal colon was fixed in 4% paraformaldehyde solution; the rest was frozen at −80 °C. Mouse blood was collected to obtain serum by centrifugation (3000× *g* rpm for 10 min at 4 °C), and the serum samples were stored at −80 °C until use. 

### 2.3. Histological Analysis

The paraformaldehyde-fixed colon samples were sequentially dehydrated with ethanol (until the final step of 100% ethanol), embedded in paraffin, sectioned, and finally stained with hematoxylin and eosin. Microscopic examination and image acquisition were performed, and the captured images were evaluated by CaseViewer software (3DHISTECH Ltd., Budapest, Hungary). The standard of histological scores was adopted from a previous publication [21]. Briefly, the scores were calculated based on: (1) loss of gut epithelial cells, (2) hyperplasia of connective tissue, (3) inflammatory cell infiltration, and (4) edema of cells.

### 2.4. Quantification of Inflammatory Cytokines in the Serum

The serum levels of several cytokines, including interleukin (IL)-1β, IL-6, IL-10, and tumor necrosis factor (TNF)-α, were quantified by ProcartaPlexTM (EPX360-26092-901, Invitrogen, Thermofisher, Watham, MA, USA). The manufacturer’s instructions were followed, and the assays were carried out in duplicate. In 96-well plates, samples were incubated for 18 h with properly diluted antibody-coupled beads. A Bio-Plex 200 system detected signals, and Bio-Plex Manager software was used to analyze the data [27].

### 2.5. Fecal DNA Extraction, Sequencing, and Analysis

Genomic DNA of fecal samples was extracted with the sodium dodecyl sulfate method [28]. The harvested DNA was checked by agarose gel electrophoresis and quantified by a Qubit^®^ 2.0 Fluorometer (Thermo Scientific). A total amount of 1 μg DNA per sample was used as input material for preparing DNA libraries. Sequencing libraries were generated using the NEBNext^®^ Ultra™ DNA Library Prep Kit for Illumina (NEB, Ipswich, MA, USA), following the manufacturer’s instructions. Index codes were added to attribute sequences to each sample. Briefly, DNA samples were fragmented by sonication to a size of ~350 bp, and then the DNA fragments were end-polished, A-tailed, and ligated with full-length adaptors for Illumina sequencing by polymerase chain reaction (PCR). The PCR products were purified by the AMPure XP system (Beckman, Brea, CA, USA) before size distribution analysis by an Agilent 2100 Bioanalyzer (Agilent, Santa Clara, CA, USA) and quantification using real-time PCR. Sequencing was performed on the Illumina NovaSeq PE150 platform at the Beijing Novogene Bioinformatics Technology Co., Ltd., China. Approximately 3 Gb of metagenomics data were generated for each sample. 

Metagenomics data were preprocessed to remove low-quality and potentially contaminating sequences. After such quality control steps, HUMAnN2 was used for taxonomic annotation.

### 2.6. Statistical Analysis

The results of body weight, immune factors, colon length/weight ratio, and histological and DAI scores were analyzed by theGraphPad Prism version 8.0.0 for Windows (GraphPad Software, San Diego, CA, USA). All statistical tests of metagenome comparison between groups were carried out on software R (version 3.5.2). Data were expressed as mean ± standard deviation (SD). Data from more than two groups were compared using Tukey’s multiple comparison tests. Adjusted *p* values below 0.05 were considered statistically significant. 

## 3. Results

### 3.1. Lactobacillus Gasseri G098 Alleviated Inflammatory Manifestations in Mice with Colitis

In this animal trial, all mice in the NC and G098 groups survived, but three mice in the DSS group died on days 8, 10, and 14, respectively, with a final survival rate of 62.5% (Figure 2A). Between days 1 and 7, the DAI scores (calculated from body weight loss, diarrhea, and bloody stool) of the DSS and G098 groups but not the NC group showed an upward trend. At day 7, the DAI scores of the DSS and G098 groups were significantly higher than that of the NC group (*p* < 0.0001), but no significant difference was observed between DSS and G098 (Figure 2B), confirming that the DSS treatment caused colitis in mice in the DSS and G098 groups. Ten days of strain intervention significantly reduced the DAI score (*p* < 0.05; Figure 2C) and weight loss (*p* < 0.05; Figure 2D) of mice in the G098 group compared with the DSS group. In addition, the colon length/weight ratio of mice in the G098 group showed numerical increase compared with those in the DSS group (*p* > 0.05; Figure 2E). 

The colon histopathological score of the DSS group was significantly higher than that of the NC group (*p* < 0.05, Figure 2F), and the histological score of G098 was significantly lower than that of the DSS group (*p* < 0.05, Figure 2F). Typical micrographs of colonic tissues of NC, DSS, and G098 groups are shown in Figure 2G. The mucosal, submucosal, muscle, and plasma membrane layers of the colon in mice of the NC group were intact, showing no obvious pathological changes. Moreover, the intestinal glands were well developed, with abundant cup cells and without obvious inflammatory cell infiltration. The gut mucosal layer of mice in the G098 group showed similar histological features, with only a small amount of inflammatory cell infiltration, but no obvious changes in local intestinal gland structures. In contrast, mice in the DSS group had detached epithelial cells in local colon mucosal layers, unremarkable intestinal gland structure, reduced cup cells, and obvious inflammatory cell infiltration in the lamina propria of mucosa. These results suggested that G098 intervention significantly prevented pathological damage of the colonic mucosa and local intestinal gland structure induced by DSS treatment.

### 3.2. Lactobacillus Gasseri G098 Intake Reversed DSS-Induced Changes in Serum Pro-/Anti-Inflammatory Cytokine Levels

The serum levels of pro-inflammatory cytokines IL-1β, IL-6, and TNF-α (Figure 3A,C) and anti-inflammatory factors IL-10 and IL-13 (Figure 3D,E) were measured at day 18. The levels of all three pro-inflammatory cytokines of the DSS group were significantly higher than that of NC group but were significantly lower in the G098 group (IL-1β: 0.71 ± 0.63 pg/mL in NC group; 10.85 ± 7.26 pg/mL in DSS group; 1.06 ± 0.76 pg/mL in G098 group; DSS vs. NC, *p* < 0.01; DSS vs. G098, *p* < 0.01; IL-6: 0.45 ± 0.49 pg/mL in NC group; 56.09 ± 48.36 pg/mL in DSS group; 8.57 ± 9.26 pg/mL in G098 group; DSS vs. NC, *p* < 0.01; DSS vs. G098, *p* < 0.05; TNF-α: 2.56 ± 0.51 pg/mL in NC group; 6.50 ± 2.28 pg/mL in DSS group; 3.22 ± 1.09 pg/mL in G098 group; DSS vs. NC, *p* < 0.001; DSS vs G098, *p* < 0.001; Figure 3A–C). The levels of IL-13 and IL-10 in the NC group were 1.84 ± 1.43 pg/mL and 9.75 ± 8.02 pg/mL, respectively; these were significantly (0.30 ± 0.25 pg/mL, *p* < 0.05; Figure 3E) and non-significantly (3.81 ± 2.99 pg/mL, *p* > 0.05; Figure 3D) lower in the DSS group. The strain intervention significantly (1.61 ± 0.88 pg/mL in G098, *p* < 0.01; Figure 3E) and non-significantly (7.32 ± 3.09 pg/mL in G098, *p* > 0.05; Figure 3D) increased the levels of IL-13 and IL-10, respectively. 

### 3.3. Effect of Lactobacillus Gasseri G098 on the Gut Microbiota Composition

The Simpson index and Shannon index are measures of microbial diversity. Mice in the DSS group had significantly lower values of Simpson and Shannon indexes compared with those in the NC group (*p* < 0.05 and *p* < 0.05, respectively), and G098 administration mitigated DSS-induced reduction in gut microbial diversity (*p* < 0.05 in both cases; Figure 4A,B). Then, the beta-diversity of mouse gut microbiota was analyzed by principal co-ordinates analysis (PCoA; Bray-Curtis distance; Figure 4C). On the PCoA plot, symbols representing the NC group clustered, while most symbols representing the DSS and G098 groups were located closely in the plot. T tests confirmed that significant difference existed between the gut microbiota structure of normal and DSS-treated mice with/without strain treatment (NC versus DSS, R^2^ = 0.3955, *p* = 0.005; NC versus G098, R^2^ = 0.2033, *p* = 0.002). However, G098 treatment could reduce this difference to some extent (R^2^ = 0.1534, *p* = 0.117), although there was a partial overlap between the G098 group and the DSS group (Figure 4C). 

The gut microbial profiles between groups were also compared. The overall fecal microbiota comprised four main phyla, including *Bacteroidetes*, *Firmicutes*, *Proteobacteria*, and *Verrucomicrobia* (Figure 4D). The DSS group had significantly more *Bacteroidetes* and *Verrucomicrobia* (*p* < 0.05 and *p* < 0.01, respectively; Figure 4E) and significantly fewer *Firmicutes* and *Proteobacteria* (*p* < 0.05 and *p* < 0.01, respectively; Figure 4E) compared with the NC group. In contrast, G098 had significantly more *Firmicutes* (*p* < 0.05; Figure 4E) but fewer *Bacteroidetes* (*p* < 0.05; Figure 4E) as compared with the DSS group. At the genus level (Figure 4F), the top three detected genera were *Muribaculaceae* (16.9% to 32.5%), *Lachnospiraceae* (5.5% to 15.7%), and *Bacteroides* (5.1% to 18.3%). To further analyze the effect of strain on the gut microbiota of DSS-induced IBD mice, bacterial genera that exhibited significant changes after G098 intervention were analyzed. Significantly, *Desulfovibrio* and *Lachnoclostridium* (*p* < 0.05 and *p* < 0.05, respectively; Figure 4G,H) but not *Muribaculaceae* (*p* < 0.05; Figure 4I) were found to be greater in the G098 group as compared with DSS group.

### 3.4. Effect of Lactobacillus Gasseri G098 on the Gut Microbiota Metabolic Pathways

The volcano plot shows the *p* value (t-test) and fold change (FC) of metabolic pathways detected in the gut microbiota (a total of 228 metabolic pathways; Figure 5A). Compared with the DSS group, five significantly increased metabolic pathways were detected in G098 group, including D-galactose degradation V (Leloir pathway), galactose degradation I (Leloir pathway), stachyose degradation, CMP-3-deoxy-D-manno-octulosonate biosynthesis I, and L-glutamate degradation V (via hydroxyglutarate) (*p* < 0.05 and FC ≥ 2 or ≤ 0.5; Figure 5A).

### 3.5. Correlation between Significant Differential Gut Microbiota with Gut Metabolic Pathways, Serum Cytokines, and DAI

We then performed a correlation analysis between the differential metabolic pathways encoded by the gut microbiota, cytokines, and DAI with significantly different bacterial genera. Several significant associations were identified (Figure 5B). *Muribaculaceae* showed significant negative correlation with D galactose degradation V (Leloir pathway; *p* < 0.01, *R* = −0.56), galactose degradation I (Leloir pathway; *p* < 0.01, *R* = −0.56), and stachyose degradation (*p* < 0.01, *R* = −0.58). In contrast, *Desulfovibrio* and *Lachnoclostridium* showed significant positive correlation with D-galactose degradation V (Leloir pathway; *p* < 0.05, *R* = 0.51; *p* < 0.01, *R* = 0.72; respectively), galactose degradation I (Leloir pathway; *p* < 0.05, *R* = 0.51; *p* < 0.01, *R* = 0.72; respectively), and stachyose degradation (*p* < 0.05, *R* = 0.54; *p* < 0.01, *R* = 0.70; respectively). Moreover, *Lachnoclostridium* also correlated positively with the pathway, L-glutamate degradation V (via hydroxyglutarate) (*p* < 0.01, *R* = 0.76), and the cytokine IL-13 (*p* < 0.01, *R* = 0.60).

## 4. Discussion

In view of the urgency in discovering unconventional but effective treatment of managing IBD, this study analyzed the beneficial effect of administering the strain *Lactobacillus gasseri* G098 on the clinical manifestations of IBDin a DSS-induced colitis mouse model. Our results supported that *Lactobacillus gasseri* G098 could effectively alleviate colitis-associated symptoms, which was accompanied by some changes in the mouse gut microbiota and encoded metabolic pathways. 

Currently, DSS is the most widely used compound to induce experimental animal models of IBD, and it is popular due to its simplicity of use, short and long intervention cycles, and reproducibility [29]. Our study successfully induced colitis in mice, simulating symptoms of IBD by administering 2.5% DSS (*w*/*v*) through drinking water for 7 days. Mice in the DSS group lost weight and had loose and bloody feces visible to the naked eye. At day 7, the DAI scores of the DSS group were significantly higher than those of the NC group (Figure 2B). The 10 d strain intervention with G098 significantly reduced the DAI scores, reversed the trend of body weight loss, and alleviated DSS-induced colitis symptoms (Figure 2C–E). As the “gold standard” to measure the severity of IBD [30], we observed and scored the colon pathology of mice by hematoxylin-eosin (HE) staining and found that G098 could significantly mitigate DSS-induced intestinal gland damage in the colon tissue, accompanied by a much reduced extent of inflammatory cell infiltration and significantly lower pathology scores compared with the DSS group (Figure 2F–G). These results together support that G098 administration was effective in alleviating IBD-associated symptoms. Importantly, there was no mortality in the G098 group throughout the experiment (versus a survival rate of 62.5% in the DSS group; Figure 2A), further confirming the protective effect of G098 intake on DSS-induced pathogenesis.

The etiology of IBD is complex, and it has been suggested that the dysregulation of the body’s immune response and cytokine disorders are the main causes of the onset of IBD [31]. The levels and functions of different cytokines are closely related to the severity of IBD, and intestinal inflammation can be alleviated by regulating the secretion of cytokines. IL-1β and IL-6, two key pro-inflammatory cytokines, are involved in a variety of autoimmune inflammatory responses and cellular activities [32,33]. It has been reported that IL-1β levels were elevated in patients with active UC, while IL-6 levels were elevated in patients with CD as compared with healthy adults; these cytokines were also reported to be positively correlated with inflammation in patients with IBD [34,35]. On the other hand, TNF-α plays an important role in the pathogenesis of IBD. It induces massive apoptosis of colonic epithelial cells and participates in the inflammatory response of the lamina propria and epithelial cell shedding, thus increasing intestinal barrier cell permeability and aggravating inflammation [36]. Currently, the development of TNF-α–targeting drugs to manage colitis is a research hotspot. For example, anti-TNF drugs, such as adalimumab, have achieved relatively satisfactory results in clinical studies, significantly reducing patients’ clinical symptoms [37,38]. Our study observed a general anti-inflammatory effect of G098 administration in DSS-treated mice, evidenced by the reduced pro-inflammatory and increased anti-inflammatory cytokine levels (Figure 3), which might be an important mechanism in protecting the mice from colitis.

The human colon hosts a large number and variety of microorganisms, which is collectively known as the gut microbiota [39], and the occurrence and remission of colitis are closely related to the gut microbiota. Numerous studies have shown that patients with IBD have reduced gut microbiota diversity, altered microbiota structure and composition, elevated levels of harmful bacteria such as *Escherichia coli*, and decreased abundance of beneficial bacteria such as *Firmicutes* and *Alistipes* [40,41]. It has been reported that the gut microbiota structure of IBD mice was disrupted and the dynamic balance between different microorganisms was disrupted, and the relative contents of *Bacteroidetes, Parabacteroides* increased significantly, while the relative contents of *Enterorhabdus* and *Prevotella* decreased significantly [42]. In line with the results of the present study, microbial metagenomic analysis found differential bacterial families between DSS and NC mice, as well as DSS versus G098 mice (Figure 4D–F). However, after the intervention of DSS, the changing trend of relative abundance of *Akkermansia muciniphila* is different from previous studies. The discrepancy might be related to factors like animal models, coexistence with other pathogenic bacteria, and host sex [43], which deserve further investigation.

As IBD is associated with the inflammatory state of the patient, and the gut microbiota plays a role in modulating the host’s immunity and inflammatory state, it is logical to regard the gut microbiota as a therapeutic target of IBD. Clinical evidence supports that modulating the gut microbiota could alleviate symptoms of colitis. Jang et al. (2018) found that *Bifidobacterium longum* LC67 and *Lactobacillus plantarum* LC27 ameliorated colitis symptoms in mice, which was accompanied by a decreased gut *Proteobacteria*/*Bacteroidetes* ratio and reduced fecal and blood lipopolysaccharide levels in mice with colitis [44]. Another study found that *Lactobacillus plantarum*-12 fed to mice with colitis could mitigate colitis symptoms, and the symptom alleviation was accompanied by an increase in gut *Lactobacillus* and a decrease in gut *Proteobacteria* [45].

Notably, our study found that G098 intake could significantly increase the alpha diversity of gut microbiota in DSS group (Figure 4A,B) and narrow the differences in the structural composition of the gut microbiota between mice DSS and NC groups (Figure 4C). The G098 treatment also significantly increased the relative abundance of gut *Firmicutes* while reducing the proportion of gut *Bacteroidetes* (Figure 4D,E). Firmicutes/Bacteroidetes ratio is a microbial dysbiosis index that could reflect the status of gut microbiota in colitis patients [46], and the increase in Firmicutes/Bacteroidetes ratio after G098 administration suggested that this strain could effectively reverse DSS-induced dysbiosis, through which the clinical manifestations of DSS-induced colitis disease were improved. At the genus level, the relative abundance of *Lachnoclostridium* of the G098 group was significantly higher than that of the DSS group (Figure 4H). *Lachnoclostridium* is a genus that produces short-chain fatty acids (SCFAs) [47], which maintain a favorable and stable colonic environment for the growth of beneficial bacteria while suppressing harmful bacterial growth in the intestine [48]. Moreover, SCFAs are closely related to intestinal inflammation, and they are involved in multiple pathways of anti-inflammation signaling [49]. Additionally, the present study found a significant positive correlation between gut *Lachnoclostridium* and the anti-inflammatory factor IL-13 (Figure 5B), indicating a potential effect of G098 administration on regulating host intestinal immune responses. On the other hand, a previous study found that *Desulfovibrio* was more abundant in the DSS group than the prebiotics intervention group [50], contrary to the current results of a lower proportion of *Desulfovibrio* in the G098 group (Figure 4G). *Desulfovibrio* can produce hydrogen sulfide (H_2_S) in the intestine, which is toxic to the intestinal epithelium and causes gastrointestinal disorders [51]. However, it has been found that H_2_S can directly promote angiogenesis and is thus beneficial to gastrointestinal ulcer healing, which might be due in part to an enhanced mucosal blood flow at the ulcer margin, facilitating ulcer repair [52]. 

Lastly, the HUMAnN2 pipeline was implemented to identify potentially functional pathways encoded by the gut microbiome, and the analysis returned a total of 228 pathways. The abundance of genes coding several pathways was higher in the G098 group compared with the DSS group, including sugar degradation-related pathways (such as stachyose degradation, D-galactose degradation V (Leloir pathway), galactose degradation I (Leloir pathway), and L-glutamate degradation V (via hydroxyglutarate); Figure 5A). Glutamate is the main raw material for producing SCFAs like acetate and butyrate, and intestinal SCFAs play an important physiological role in energy supply to intestinal epithelial cells, electrolyte homeostasis, maintenance of intestinal mucosal barrier, and regulation of immune and antitumor effects [53]. The metabolic pathway of L-glutamate degradation V (via hydroxyglutarate) was significantly and positively correlated with *Lachnoclostridium*, which is a bacterial family known to produced SCFAs [54]. Thus, G098 may stimulate the growth of the portion of gut microbiota that encoded the aforementioned pathways and were responsible for a number of physiological functions, including SCFA production and strengthening of the intestinal mucosal barrier, alleviating intestinal inflammation [55].

Therefore, our data showed that *Lactobacillus gasseri* G098 treatment could mitigate symptoms of DSS-induced inflammatory bowel disease in mice. In the future, we need to further explore the specific mechanism of Lactobacillus gasseri G098 in relieving colitis and continue to carry out relevant longitudinal studies to provide more evidence to apply *Lactobacillus gasseri* G098 treatment for the prevention and treatment of IBDs. 

## 5. Conclusions

In conclusion, *Lactobacillus gasseri* G098 could alleviate DSS-induced colitis in mice by reducing mucosal damage in colonic tissues, modulating immune responses, restoring gut microbiota diversity, and increasing gut microbiota stability. Our results suggested that *Lactobacillus gasseri* G098 should be further evaluated for its clinical efficacy in managing IBD. However, the specific mechanism of *Lactobacillus gasseri* G098 mitigation in experimental colitis requires further study.

## Figures and Tables

**Figure 1 nutrients-14-03745-f001:**
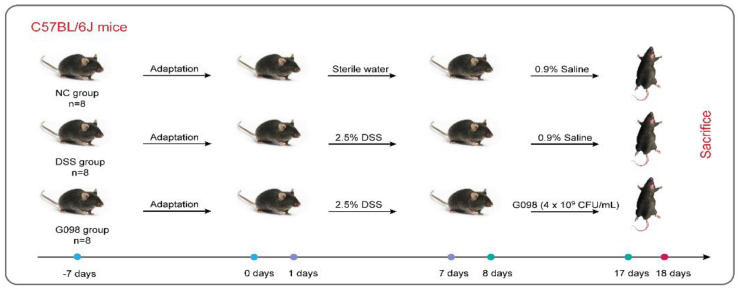
Experimental design. “NC”, “DSS”, and “G098” groups represent normal control, dextran sodium sulfate-induced, and *Lactobacillus gasseri* G098 groups, respectively.

**Figure 2 nutrients-14-03745-f002:**
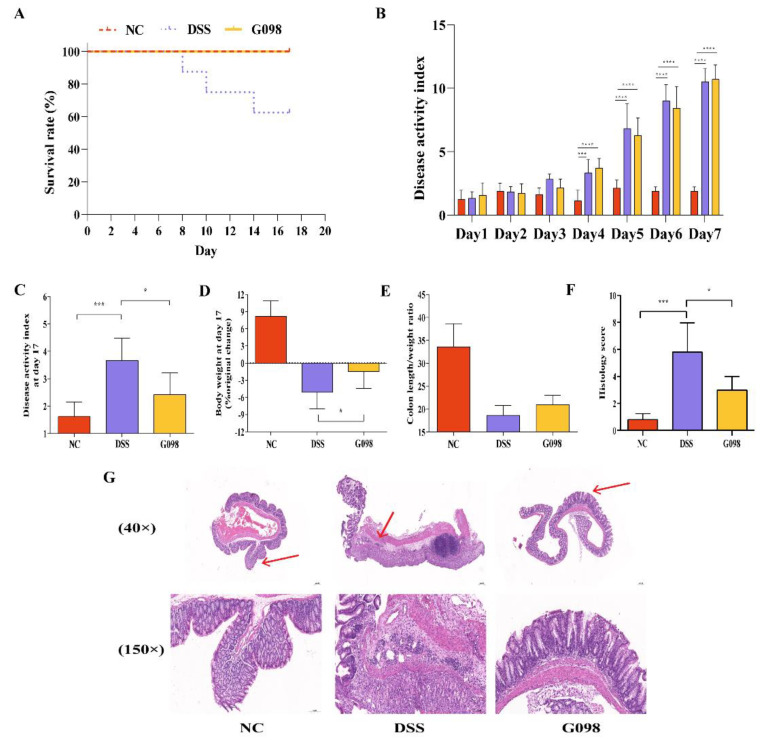
Differences in inflammatory manifestations in normal control (NC), dextran sodium sulfate (DSS)-induced, and *Lactobacillus gasseri* G098 (G098) groups. (**A**) Survival rate. (**B**) Changes in disease activity index. (**C**) Disease activity index and (**D**) body weight change at day 17 relative to baseline. (**E**) Colon length/weight ratio and (**F**) histology score. (**G**) Hematoxylin and eosin staining microscopic images (40× and 150× magnification) of colon tissues; microscopic images (150× magnification) are a magnified image of the area indicated by the red arrow. Error bars represent standard deviation of the mean. Tukey’s multiple comparison tests were used for statistical analysis (*, *p* < 0.05; ***, *p* < 0.001; ****, *p* < 0.0001).

**Figure 3 nutrients-14-03745-f003:**
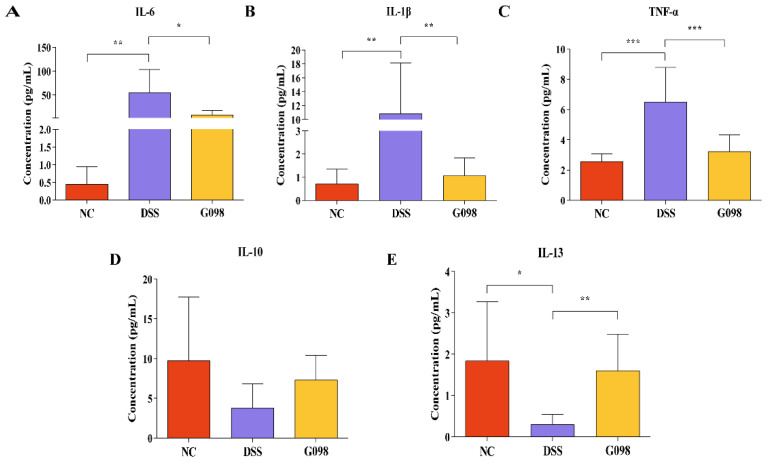
Differences in serum cytokine levels in normal control (NC), dextran sodium sulfate (DSS)-induced, and *Lactobacillus gasseri* G098 (G098) groups. Levels of (**A**) interleukin (IL)-6; (**B**) IL-1β; (**C**) Tumor necrosis factor (TNF)-α; (**D**) IL-10; and (**E**) IL-13. Error bars represent standard deviation of the mean. Tukey’s multiple comparison tests were used to evaluate differences between groups (*, *p* < 0.05; **, *p* < 0.01; ***, *p* < 0.001).

**Figure 4 nutrients-14-03745-f004:**
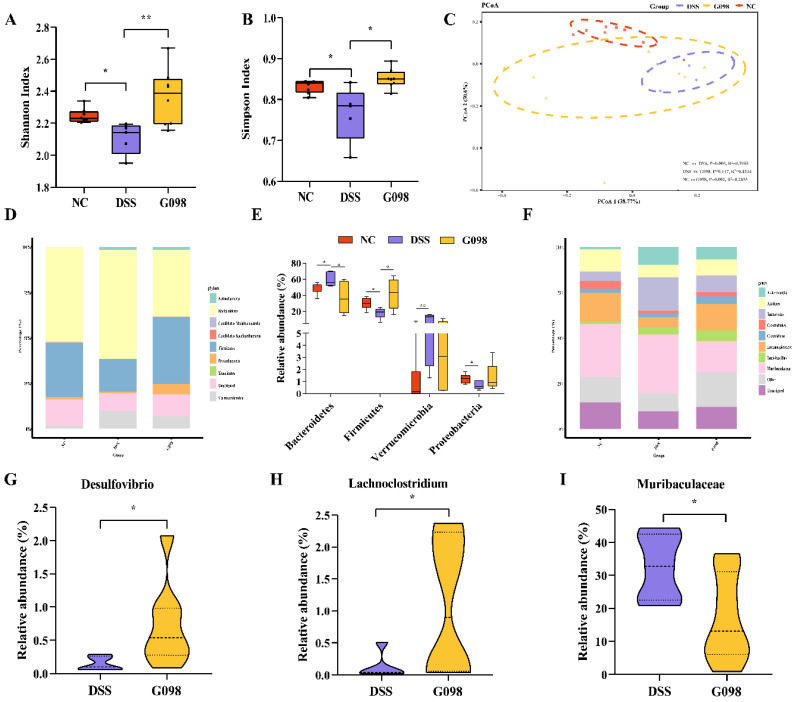
The taxonomic gut microbiota of three groups (normal control (NC), dextran sodium sulfate (DSS)-induced, and *Lactobacillus gasseri* G098 (G098) groups). (**A**) Shannon index. (**B**) Simpson index. (**C**) Principal coordinate analysis (PCoA; Bray-Curtis distance) score plot of mouse gut microbiota. (**D**) Phylum-level gut microbiota profile. (**E**) The relative abundance of *Bacteroidetes*, *Firmicutes*, *Proteobacteria*, and *Verrucomicrobia*. (**F**) Genus-level gut microbiota profile. (**G**–**I**) Violin plots showing relative abundances of *Desulfovibrio*, *Lachnoclostridium*, and *Muribaculaceae*. Error bars represent standard deviation of the mean. Tukey’s multiple comparison tests were used for statistical analysis (*, *p* < 0.05; **, *p* < 0.01).

**Figure 5 nutrients-14-03745-f005:**
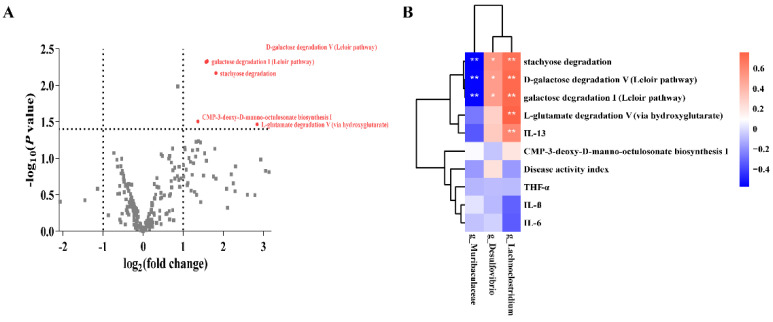
Differential taxa, metabolic pathways, and correlation analysis. (**A**) Volcano map showing differential metabolic pathways between the gut microbiota of dextran sodium sulfate (DSS)-induced and *Lactobacillus gasseri* G098 (G098) groups (cut-off: *p* < 0.05 and FC ≥ 2 or ≤ 0.5). Each dot represents one detected pathway. Differential metabolic pathways are shown in red and are annotated. (**B**) Spearman’s correlation heatmap showing association between differential taxa, metabolic pathways, serum cytokine levels, and disease activity index. * and ** represent *p* < 0.05 and *p* < 0.01, respectively. The color scale represents *r*-value.

## Data Availability

Not applicable.

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
