# Peer review of "The *Lactobacillus gasseri* G098 Strain Mitigates Symptoms of DSS-Induced Inflammatory Bowel Disease in Mice"

_nutrients, 2022, doi:10.3390/nu14183745_

Round 1

Reviewer 1 Report

The paper presented to me for review is another one addressing the problems of the beneficial effects of probiotic bacteria on colitis. The authors conducted an experimental study on mice. The experimental model of the study consisted of animals with experimentally induced colitis by administration of DDS. The research methods used as well as the experimental design do not give me any objections. 

In the introduction, the authors clearly and compactly presented the issues related to the prepared work.  

In my opinion, the work deals with the theoretically important issue of the effect of bacteria of Lactobacillus griseus strain G098 on inflammatory processes taking place in the large intestine. In my opinion, the authors developed the experimental model without objection. 

They presented the plan and results of their research in a clear and transparent way in the form of 5 figures. 

The discussion of the obtained research results is interesting and comprehensive. 

Conclusions from the obtained research results should be presented by the authors in a separate chapter. 

In the publication they included 55 items of current literature. The scientific literature is cited in an appropriate manner. I propose to accept the paper for publication after minor correction

Reviewer 2 Report

The authors have demonstrated the efficacy of Lactobacillus griseus G098 oral treatment in reducing and reversing the inflammatory effects of DSS in mice. The effects have been demonstrated by survival, Disease Activity Index, histopathology and by measuring the inflammatory cytokines. The effects of G098 on the intestinal microbiota and on different intestinal metabolic pathways have also been studied. Overall, the manuscript presents a well designed intervention experiment with a potential interest in the treatment of inflammatory bowel diseases.

My few comments/questions are the following:

1) Microbial nomenclature. The taxonomy of the genus Lactobacillus has been under a major revision and the names of most species have been changed after splitting the former Lactobacillus genus into more than 20 novel genera. The authors should check that their nomenclature is up to date.

2) Materials and methods, Section 2.1. Animals and experimental strain. The chapter starts with a sentence: "All subjects gave their informed consent for inclusion before they participated in the study." Because all the subjects were mice, they hardly were able to give their informed consent.

3) Were there any data of the survival and/or intestinal colonization of G098 in the treated mice?

4) Why were no faecal samples collected before the start of the trial to map te baseline situation?
